

# Formatting Open Science: agilely creating multiple document formats for academic manuscripts with Pandoc Scholar

Albert Krewinkel[1] and Robert Winkler[2]

[1] Pandoc Development Team, Berlin, Germany
[2] Department of Biotechnology and Biochemistry, CINVESTAV Unidad Irapuato, Mexico

Corresponding author
Robert Winkler,
robert.winkler@cinvestav.mx

## ABSTRACT

The timely publication of scientific results is essential for dynamic advances in science. The ubiquitous availability of computers which are connected to a global network made the rapid and low-cost distribution of information through electronic channels possible. New concepts, such as Open Access publishing and preprint servers are currently changing the traditional print media business towards a community-driven peer production. However, the cost of scientific literature generation, which is either charged to readers, authors or sponsors, is still high. The main active participants in the authoring and evaluation of scientific manuscripts are volunteers, and the cost for online publishing infrastructure is close to negligible. A major time and cost factor is the formatting of manuscripts in the production stage. In this article we demonstrate the feasibility of writing scientific manuscripts in plain markdown (MD) text files, which can be easily converted into common publication formats, such as PDF, HTML or EPUB, using Pandoc. The simple syntax of Markdown assures the long-term readability of raw files and the development of software and workflows. We show the implementation of typical elements of scientific manuscripts—formulas, tables, code blocks and citations—and present tools for editing, collaborative writing and version control. We give an example on how to prepare a manuscript with distinct output formats, a DOCX file for submission to a journal, and a LATEX/PDF version for deposition as a PeerJ preprint. Further, we implemented new features for supporting 'semantic web' applications, such as the 'journal article tag suite'—JATS, and the 'citation typing ontology'—CiTO standard. Reducing the work spent on manuscript formatting translates directly to time and cost savings for writers, publishers, readers and sponsors. Therefore, the adoption of the MD format contributes to the agile production of open science literature. Pandoc Scholar is freely available from https://github.com/pandoc-scholar.

## INTRODUCTION

Agile development of science depends on the continuous exchange of information between researchers (*Woelfle, Olliaro & Todd, 2011*). In the past, physical copies of scientific works had to be produced and distributed. Therefore, publishers needed to invest considerable

resources for typesetting and printing. Since the journals were mainly financed by their subscribers, their editors not only had to decide on the scientific quality of a submitted manuscript, but also on the potential interest to their readers. The availability of globally connected computers enabled the rapid exchange of information at low cost. Yochai Benkler *(2006)* predicts important changes in the information production economy, which are based on three observations:

1. A nonmarket motivation in areas such as education, arts, science, politics and theology.
2. The actual rise of nonmarket production, made possible through networked individuals and coordinate effects.
3. The emergence of large-scale peer production; for example, of software and encyclopedias.

Immaterial goods such as knowledge and culture are not lost when consumed or shared—they are 'nonrival'—, and they enable a networked information economy, which is not commercially driven (*Benkler, 2006*).

## Preprints and e-prints

In some areas of science a preprint culture, i.e., a paper-based exchange system of research ideas and results, already existed when Paul Ginsparg in 1991 initiated a server for the distribution of electronic preprints—'e-prints'—about high-energy particle theory at the Los Alamos National Laboratory (LANL), USA (*Ginsparg, 1994*). Later, the LANL server moved with Ginsparg to Cornell University, USA, and was renamed as arXiv (*Butler, 2001*). Currently, arXiv (https://arxiv.org/) publishes e-prints related to physics, mathematics, computer science, quantitative biology, quantitative finance and statistics. Just a few years after the start of the first preprint servers, their important contribution to scientific communication was evident (*Ginsparg, 1994*; *Youngen, 1998*; *Brown, 2001*). In 2014, arXiv reached the impressive number of 1 million e-prints (*Van Noorden, 2014*).

In more conservative areas, such as chemistry and biology, accepting the publishing prior peer-review took more time (*Brown, 2003*). A preprint server for life sciences (http://biorxiv.org/) was launched by the Cold Spring Habor Laboratory, USA, in 2013 (*Callaway, 2013*). *PeerJ preprints* (https://peerj.com/preprints/), started in the same year, accepts manuscripts from biological sciences, medical sciences, health sciences and computer sciences.

The terms 'preprints' and 'e-prints' are used synonymously, since the physical distribution of preprints has become obsolete. A major drawback of preprint publishing are the sometimes restrictive policies of scientific publishers. The SHERPA/RoMEO project informs about copyright policies and self-archiving options of individual publishers (http://www.sherpa.ac.uk/romeo/).

## Open Access

The term '*Open Access*' (OA) was introduced 2002 by the Budapest Open Access Initiative and was defined as:

*"Barrier-free access to online works and other resources. OA literature is digital, online, free of charge (gratis OA), and free of needless copyright and licensing restrictions (libre OA)."* (*Suber, 2012*)

Frustrated by the difficulty to access even digitized scientific literature, three scientists founded the *Public Library of Science (PLoS)*. In 2003, *PLoS Biology* was published as the first fully Open Access journal for biology (*Brown, Eisen & Varmus, 2003*; *Eisen, 2003*).

Thanks to the great success of OA publishing, many conventional print publishers now offer a so-called 'Open Access option', i.e., to make accepted articles free to read for an additional payment by the authors. The copyright in these hybrid models might remain with the publisher, whilst fully OA usually provide a liberal license, such as the Creative Commons Attribution 4.0 International (CC BY 4.0, https://creativecommons.org/licenses/by/4.0/).

OA literature is only one component of a more general *open* philosophy, which also includes the access to scholarships, software, and data (*Willinsky, 2005*). Interestingly, there are several different 'schools of thought' on how to understand and define *Open Science*, as well the position that any science is open by definition, because of its objective to make generated knowledge public (*Fecher & Friesike, 2014*).

## Cost of journal article production

In a recent study, the article processing charges (APCs) for research intensive universities in the USA and Canada were estimated to be about 1,800 USD for fully OA journals and 3,000 USD for hybrid OA journals (*Solomon & Björk, 2016*). PeerJ (https://peerj.com/), an OA journal for biological and computer sciences launched in 2013, drastically reduced the publishing cost, offering its members a life-time publishing plan for a small registration fee (*Van Noorden, 2012*); alternatively the authors can choose to pay an APC of $1,095 USD, which may be cheaper, if multiple co-authors participate.

Examples such as the *Journal of Statistical Software* (*JSS*, https://www.jstatsoft.org/) and *eLife* (https://elifesciences.org/) demonstrate the possibility of completely community-supported OA publications. Figure 1 compares the APCs of different OA publishing business models.

*JSS* and *eLife* are peer-reviewed and indexed by Thomson Reuters. Both journals are located in the Q1 quality quartile in all their registered subject categories of the Scimago Journal & Country Rank (http://www.scimagojr.com/), demonstrating that high-quality publications can be produced without charging the scientific authors or readers.

In 2009, a study was carried out concerning the ''*Economic Implications of Alternative Scholarly Publishing Models*'', which demonstrates an overall societal benefit by using OA publishing models (*Houghton et al., 2009*). In the same report, the real publication costs are evaluated. The relative costs of an article for the publisher are presented in Fig. 2.

Conventional publishers justify their high subscription or APC prices with the added value; for example, journalism (stated in the graphics as 'non-article processing'). However, stakeholder profits, which could be as high as 50%, also must be considered, and are withdrawn from the science budget (*Van Noorden, 2013*).

Generally, the production costs of an article could be roughly divided into commercial and academic/technical costs (Fig. 2). For nonmarket production, the commercial costs such as margins/profits, management etc. can be drastically reduced. Hardware and services for hosting an editorial system, such as Open Journal Systems of the Public Knowledge Project (https://pkp.sfu.ca/ojs/) can be provided by public institutions. Employed scholars

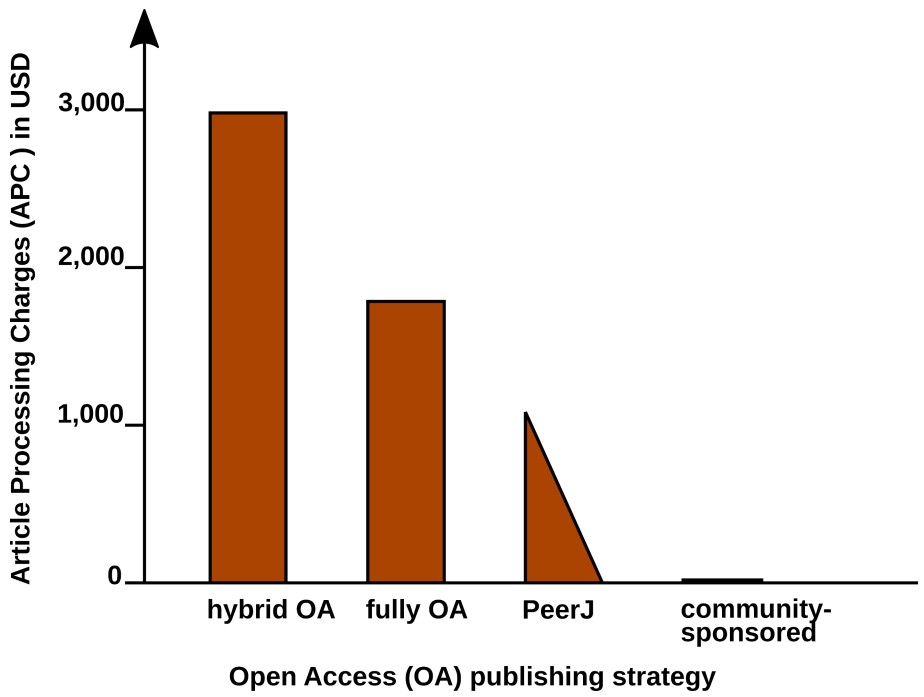

**Figure 1** Article Processing Charge (APCs) that authors have to pay for with different Open Access (OA) publishing models. Data from *Solomon & Björk (2016)* and journal web-pages.

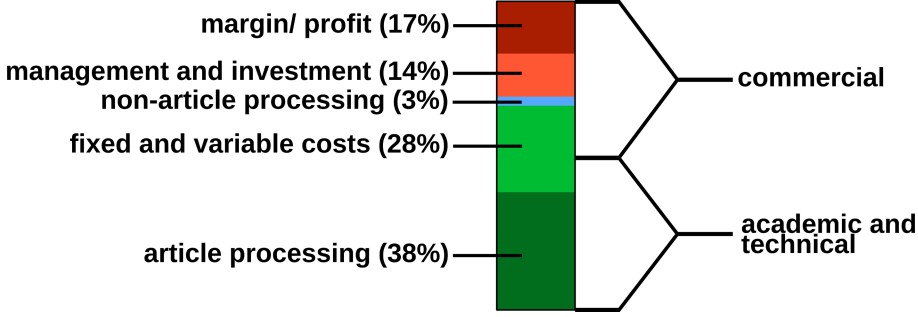

**Figure 2** Estimated publishing cost for a 'hybrid' journal (conventional with Open Access option). Data from *Houghton et al. (2009)*.

can perform editor and reviewer activities without additional cost for the journals. Nevertheless, 'article processing', which includes the manuscript handling during peer review and production represents the most expensive part.

Therefore, we investigated a strategy for the efficient formatting of scientific manuscripts.

## Current standard publishing formats

Generally speaking, a scientific manuscript is composed of contents and formatting. While the content, i.e., text, figures, tables, citations etc., may remain the same between different publishing forms and journal styles, the formatting can be very different. Most publishers require the formatting of submitted manuscripts in a certain format. Ignoring this **Guide**

**Table 1** Current standard formats for scientific publishing.

| Type | Description | Use | Syntax | Reference |
|---|---|---|---|---|
| DOCX | Office Open XML | WYSIWYG editing | XML, ZIP | *Ngo (2006)* |
| ODT | OpenDocument | WYSIWYG editing | XML, ZIP | *Brauer et al. (2005)* |
| PDF | Portable document | Print replacement | PDF | *International Organization for Standardizationd (2013)* |
| EPUB | Electronic publishing | e-books | HTML5, ZIP | *Eikebrokk, Dahl & Kessel (2014)* |
| JATS | Journal article tag suite | Journal publishing | XML | *National Information Standards Organization (2012)* |
| LATEX | Typesetting system | High-quality print | TEX | *Lamport (1994)* |
| HTML | Hypertext markup | Websites | (X)HTML | *Raggett, Le Hors & Jacobs (1999)* and *Hickson et al. (2014)* |
| MD | Markdown | Lightweight markup | Plain text MD | *Ovadia (2014)* and *Leonard (2016)* |

**Table 2** Examples for formatting elements and their implementations in different markup languages.

| Element | Markdown | LATEX | HTML |
|---|---|---|---|
| **Structure** | | | |
| Section | `# Intro` | `\section{Intro}` | `<h1>Intro</h1>` |
| Subsection | `## History` | `\subsection{History}` | `<h2>History</h2>` |
| **Text style** | | | |
| Bold | `**text**` | `\textbf{text}` | `<b>text</b>` |
| Italics | `*text*` | `\textit{text}` | `<i>text</i>` |
| **Links** | | | |
| HTTP link | `<https://arxiv.org>` | `\usepackage{url}\url{https://arxiv.org}` | `<a href="https://arxiv.org"></a>` |

**for Authors**, (for example, by submitting a manuscript with a different reference style), gives a negative impression with a journal's editorial staff. Manuscripts which are too carelessly prepared can even provoke a straight 'desk-reject' (*Volmer & Stokes, 2016*).

Currently DOC(X), LATEX and/ or PDF file formats are the most frequently used formats for journal submission platforms. However, even if the content of a submitted manuscript might be accepted during the peer review 'as is', the format still needs to be adjusted to the particular publication style in the production stage. For the electronic distribution and archiving of scientific works, which is gaining more and more importance, additional formats (EPUB, (X)HTML, JATS) need to be generated. Table 1 lists the file formats which are currently the most relevant ones for scientific publishing.

Although the content elements of documents, such as title, author, abstract, text, figures, tables, etc., remain the same, the syntax of the file formats is rather different. Table 2 demonstrates some simple examples of differences in different markup languages.

Documents with the commonly used Office Open XML (DOCX Microsoft Word files) and OpenDocument (ODT LibreOffice) file formats can be opened in a standard text editor after unzipping. However, content and formatting information is distributed into various folders and files. Practically speaking, those file formats require the use of special word processing software.

From a writer's perspective, the use of *What You See Is What You Get (WYSIWYG)* programs such as Microsoft Word, WPS Office or LibreOffice might be convenient, because

the formatting of the document is directly visible. But the complicated syntax specifications often result in problems when using different software versions and for collaborative writing. Simple conversions between file formats can be difficult or impossible. In a worst-case scenario, 'old' files cannot be opened any more for lack of compatible software.

In some parts of the scientific community therefore LATEX, a typesetting program in plain text format, is very popular. With LATEX, documents with highest typographic quality can be produced. However, the source files are cluttered with LATEX commands and the source text can be complicated to read. Causes of compilation errors in LATEX are sometimes difficult to find. Therefore, LATEX is not very user friendly, especially for casual writers or beginners.

In academic publishing, it is additionally desirable to create different output formats from the same source text:

- For the publishing of a book, with a print version in PDF and an electronic version in EPUB.
- For the distribution of a seminar script, with an online version in HTML and a print version in PDF.
- For submitting a journal manuscript for peer-review in DOCX, as well as a preprint version with another journal style in PDF.
- For archiving and exchanging article data using the Journal Article Tag Suite (JATS) (*National Information Standards Organization, 2012*), a standardized format developed by the National Library of Medicine (NLM).

Some of the tasks can be performed with LATEX, but an integrated solution remains a challenge. Several programs for the conversion between documents formats exist, such as the e-book library program calibre (http://calibre-ebook.com/). But the results of such conversions are often not satisfactory and require substantial manual corrections.

Therefore, we were looking for a solution that enables the creation of scientific manuscripts in a simple format, with the subsequent generation of multiple output formats. The need for hybrid publishing has been recognized outside of science (*Kielhorn, 2011*), but the requirements specific to scientific publishing have not been addressed so far. Therefore, we investigated the possibility to generate multiple publication formats from a simple manuscript source file.

## CONCEPTS OF MARKDOWN AND PANDOC

Markdown was originally developed by John Gruber in collaboration with Aaron Swartz, with the goal to simplify the writing of HTML documents (http://daringfireball.net/projects/markdown/). Instead of coding a file in HTML syntax, the content of a document is written in plain text and annotated with simple tags which define the formatting. Subsequently, the Markdown (MD) files are parsed to generate the final HTML document. With this concept, the source file remains easily readable and the author can focus on the contents rather than formatting. Despite its original focus on the web, the MD format has been proven to be well suited for academic writing (*Ovadia, 2014*). In particular, Pandoc-flavored MD (http://pandoc.org/) adds several extensions which facilitate the

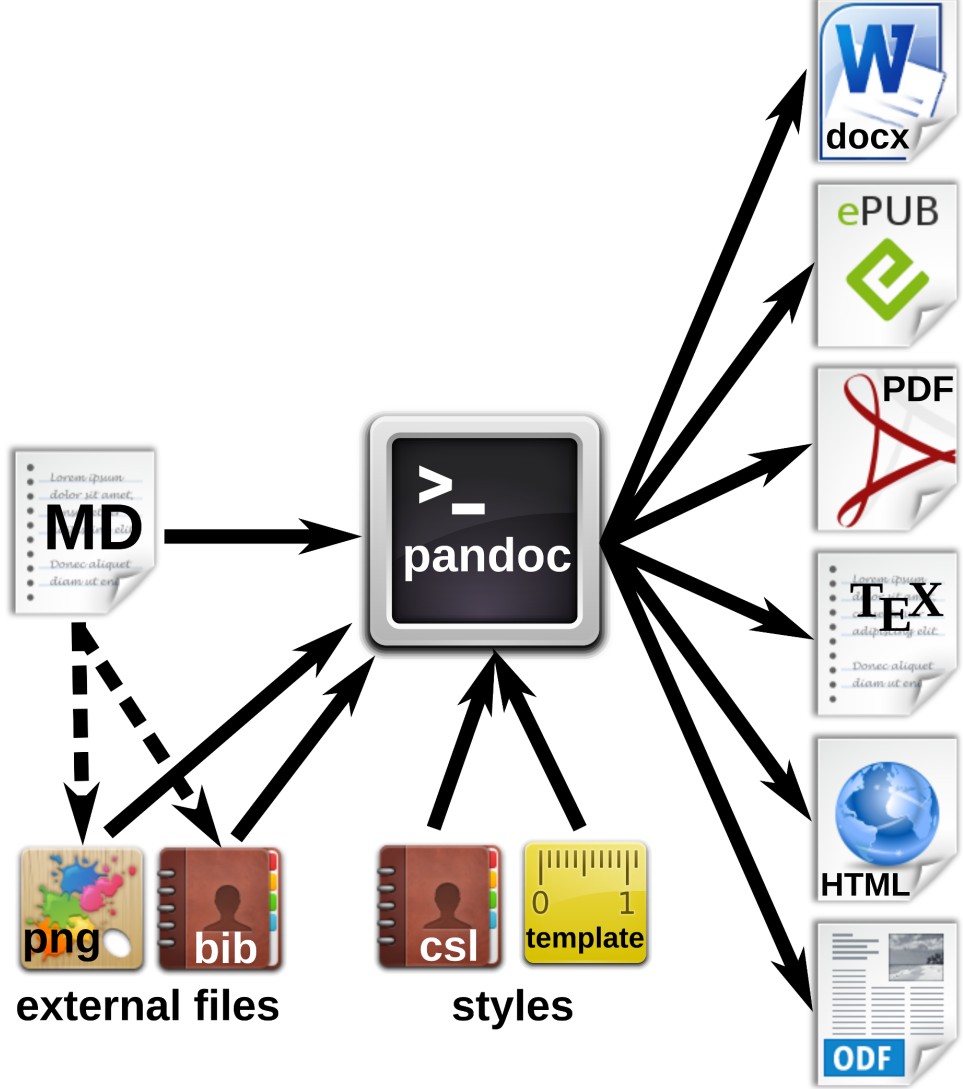

**Figure 3** **Workfow for the generation of multiple document formats with Pandoc.** The markdown (MD) file contains the manuscript text with formatting tags, and can also refer to external files such as images or reference databases. The Pandoc processor converts the MD file to the desired output formats. Documents, citations etc. can be defined in style files or templates.

authoring of academic documents and their conversion into multiple output formats. Table 2 demonstrates the simplicity of MD compared to other markup languages. Figure 3 illustrates the generation of various formatted documents from a manuscript in Pandoc MD. Some relevant functions for scientific texts are explained below in more detail.

## MARKDOWN EDITORS AND ONLINE EDITING

The usability of a text editor is important for the author, either writing alone or with several co-authors. In this section we present software and strategies for different scenarios. Figure 4 summarizes various options for local or networked editing of MD files.

**cloud drive**                    **git server (version control)**

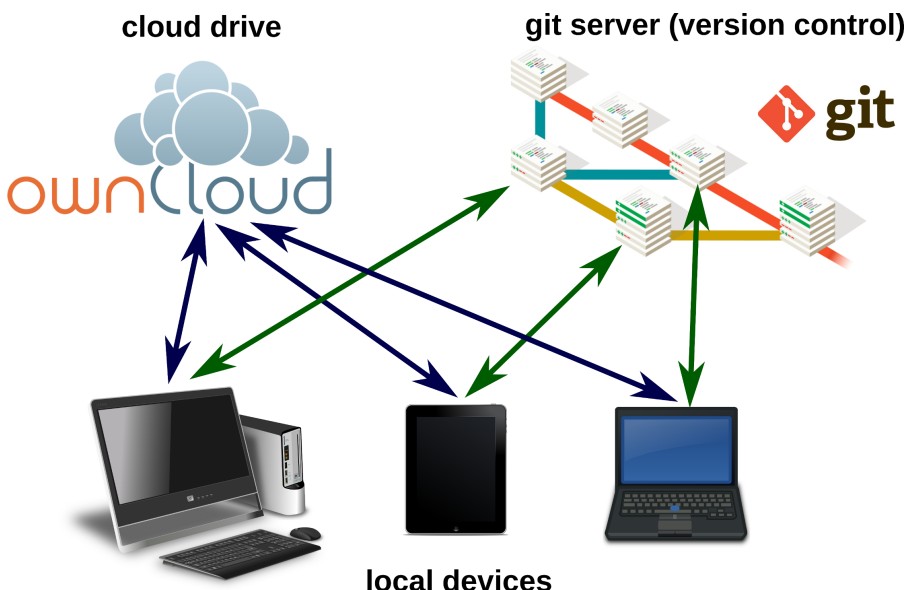

**local devices**

**Figure 4  Markdown files can be edited on local devices or on cloud drives.** A local or remote Git repository enables advanced advanced version control.

## Markdown editors

Due to MD's simple syntax, basically any text editor is suitable for editing markdown files. The formatting tags are written in plain text and are easy to remember. Therefore, the author is not distracted by looking around for layout options with the mouse. For several popular text editors, such as vim (http://www.vim.org/), GNU Emacs (https://www.gnu.org/software/emacs/), atom (https://atom.io/) or geany (http://www.geany.org/), plugins provide additional functionality for markdown editing; for example, syntax highlighting, command helpers, live preview or structure browsing.

Various dedicated markdown editors have been published as well. Many of those are cross-platform compatible, such as Abricotine (http://abricotine.brrd.fr/), ghostwriter (https://github.com/wereturtle/ghostwriter) and CuteMarkEd (https://cloose.github.io/CuteMarkEd/).

The lightweight format is also ideal for writing on mobile devices. Numerous applications are available on the App stores for Android and iOS systems. The programs Swype and Dragon (http://www.nuance.com/) facilitate the input of text on such devices by guessing words from gestures and speech recognition (dictation).

Figure 5 shows the editing of a markdown file, using the cross-platform editor Atom with several markdown plugins.

## Online editing and collaborative writing

Storing manuscripts on network drives (*The Cloud*) has become popular for several reasons:

- Protection against data loss.
- Synchronization of documents between several devices.
- Collaborative editing options.

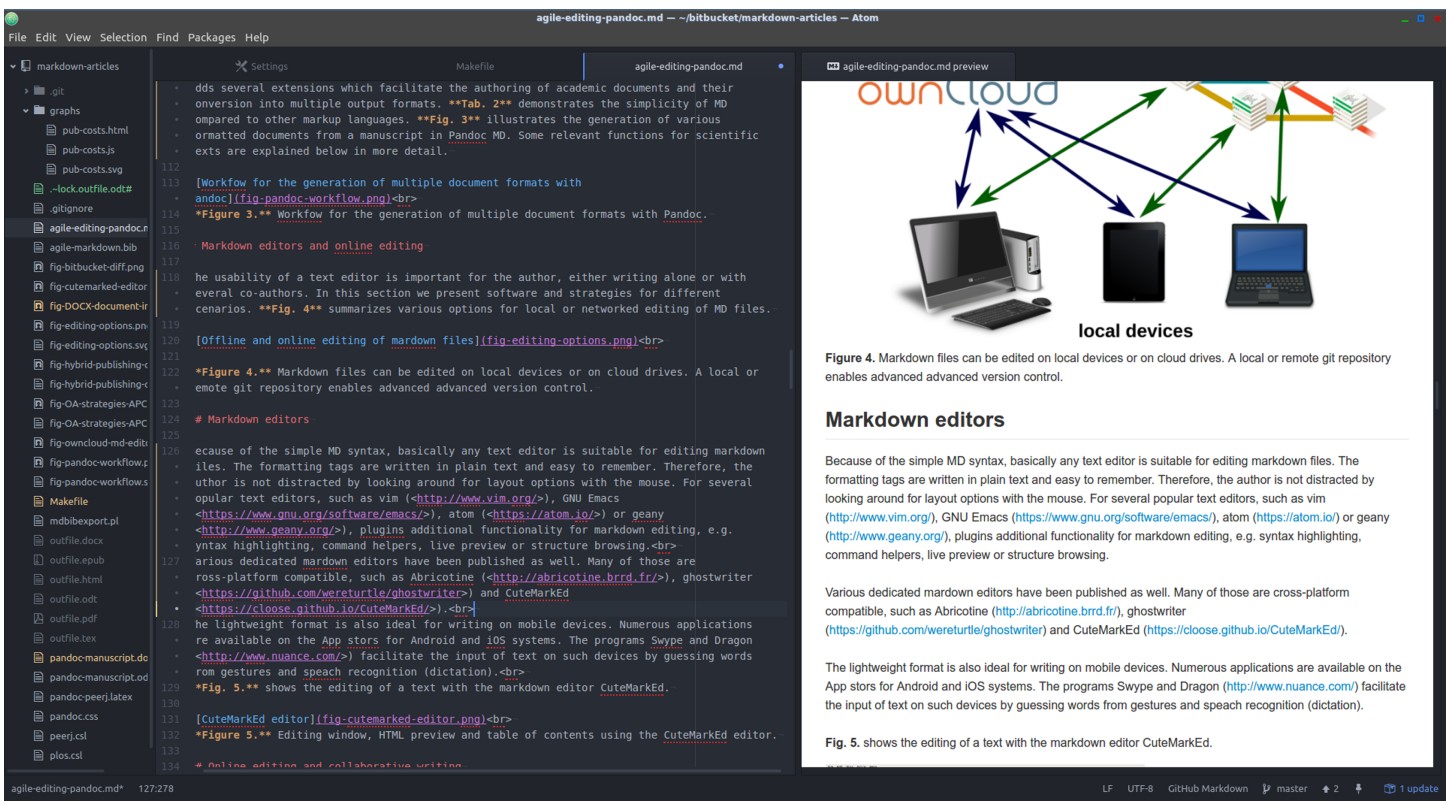

**Figure 5**  Document directory tree, editing window and HTML preview using the Atom editor.

Markdown files on a Google Drive (https://drive.google.com) for instance can be edited online with StackEdit (https://stackedit.io). Figure 6 demonstrates the online editing of a markdown file on an ownCloud (https://owncloud.com/) installation. OwnCloud is an Open Source software platform, which allows the set-up of a file server on personal webspace. The functionality of an ownCloud installation can be enhanced by installing plugins.

Even mathematical formulas are rendered correctly in the HTML live preview window of the ownCloud markdown plugin (Fig. 6).

The collaboration and authoring platform Authorea (https://www.authorea.com/) also supports markdown as one of multiple possible input formats. This can be beneficial for collaborations in which one or more authors are not familiar with markdown syntax.

## Document versioning and change control

Programmers, especially when working in distributed teams, rely on version control systems to manage changes of code. Currently, Git (https://git-scm.com/), which is also used for the development of the Linux kernel, is one of the most employed software solutions for versioning. Git allows the parallel work of collaborators and has an efficient merging and conflict resolution system. A Git repository may be used by a single local author to keep track of changes, or by a team with a remote repository; for example, on Github (https://github.com/) or Bitbucket (https://bitbucket.org/). Because of the plain text format of markdown, Git can be used for version control and distributed writing. For

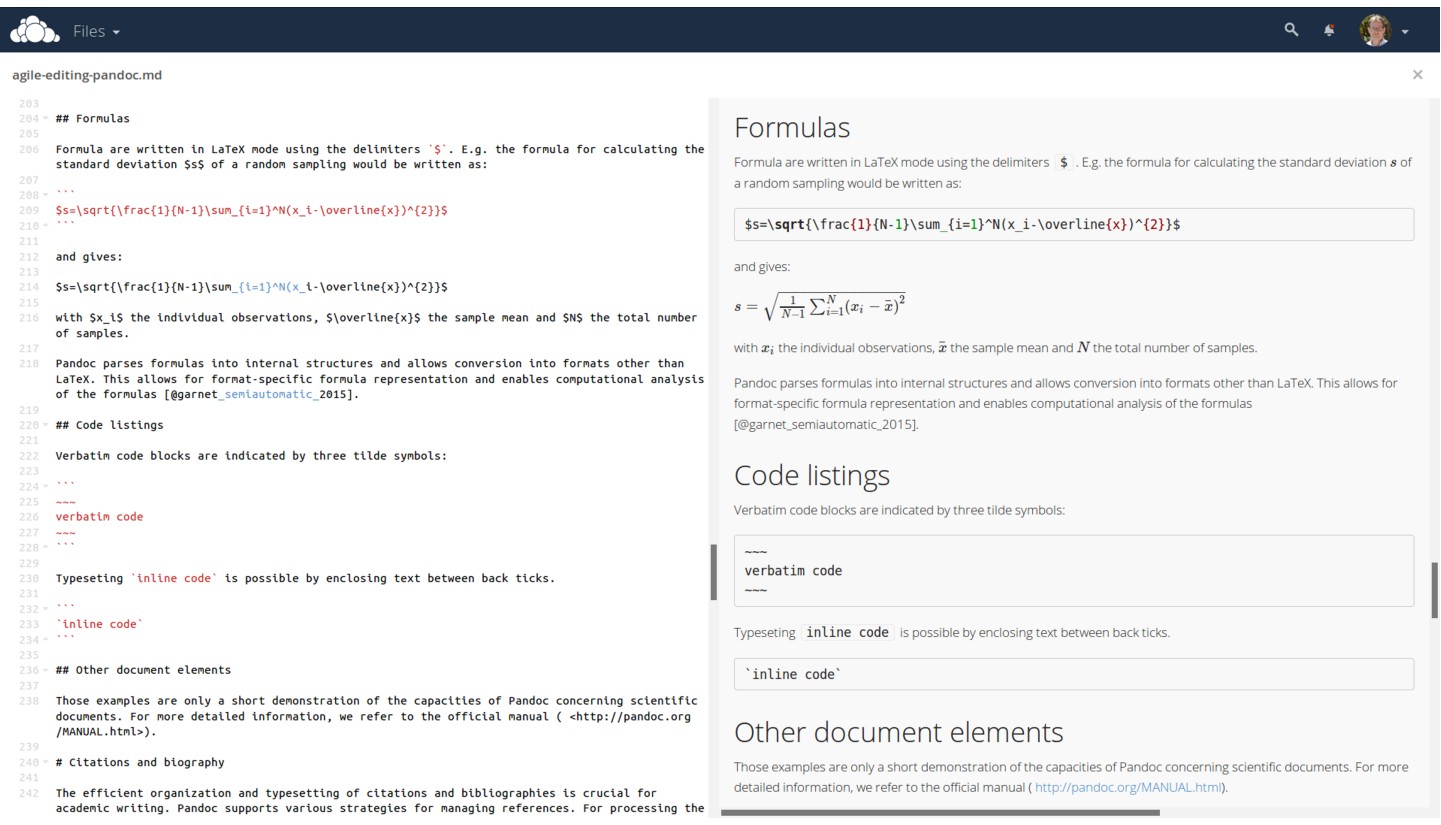

**Figure 6** Direct online editing of this manuscript with live preview using the ownCloud Markdown Editor plugin by Robin Appelman.

the writing of the present article, the co-authors (Germany and Mexico) used a remote Git repository on Bitbucket. The plain text syntax of markdown facilitates the visualization of differences of document versions, as shown in Fig. 7.

# PANDOC MARKDOWN FOR SCIENTIFIC TEXTS

In the following section, we demonstrate the potential for typesetting scientific manuscripts with Pandoc using examples for typical document elements, such as tables, figures, formulas, code listings and references. A brief introduction is given by *Dominici (2014)*. The complete Pandoc User's Manual is available at http://pandoc.org/MANUAL.html.

## Tables

There are several options to write tables in markdown. The most flexible alternative—which was also used for this article—are pipe tables. The contents of different cells are separated by pipe symbols (|):

```
Left | Center | Right | Default
:-----|:------:|------:|---------
LLL   | CCC     | RRR    | DDD
```

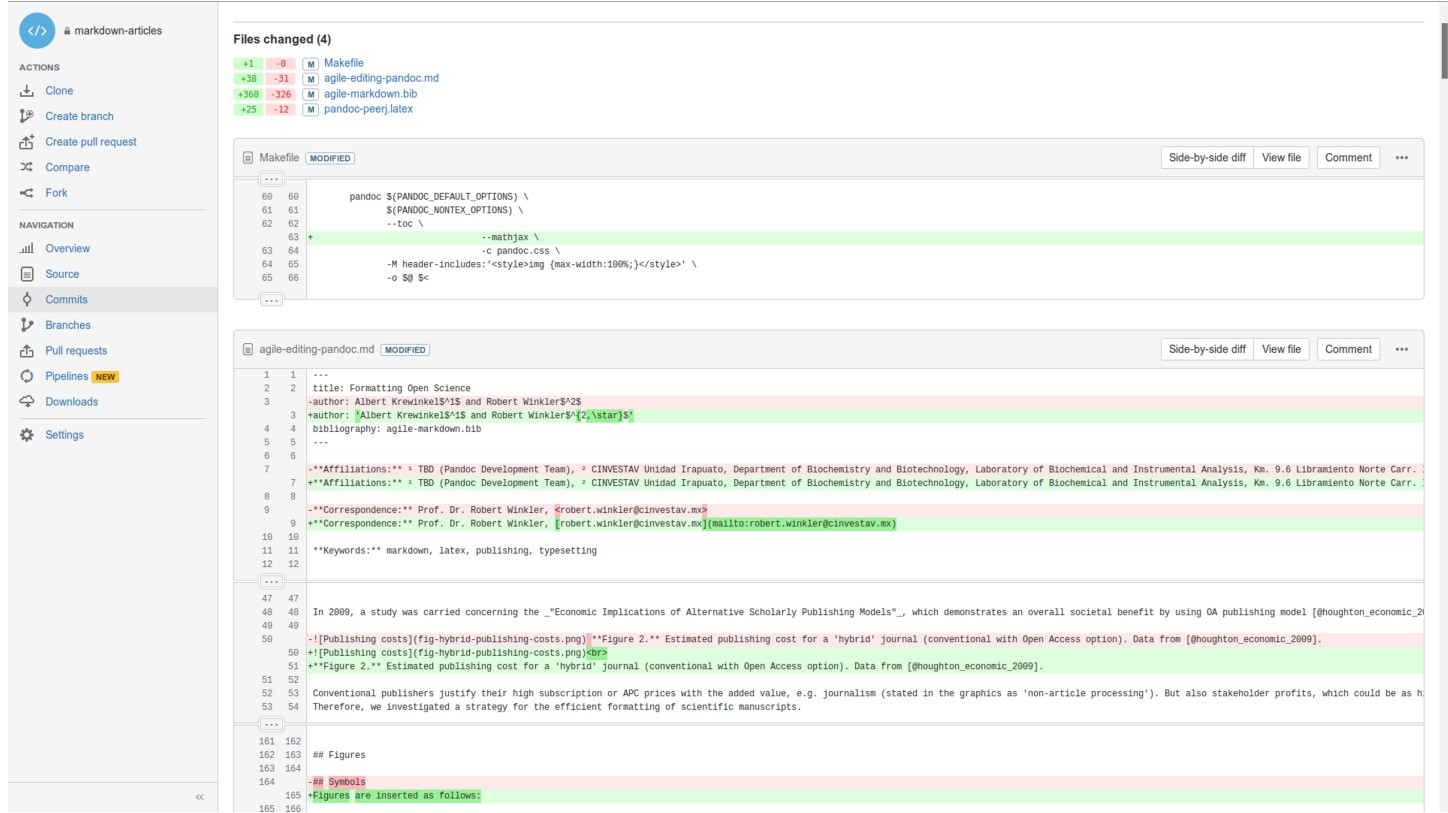

**Figure 7** Version control and collaborative editing using a Git repository on Bitbucket.

gives

| Left | Center | Right | Default |
|------|--------|-------|---------|
| LLL  | CCC    | RRR   | DDD     |

The headings and the alignment of the cells are given in the first two lines. The cell width is variable. The Pandoc parameter `--columns=NUM` can be used to define the length of lines in characters. If contents do not fit, they will be wrapped.

Complex tables (for example, tables featuring multiple headers or those containing cells spanning multiple rows or columns), are currently not representable in Markdown format. However, it is possible to embed LATEX and HTML tables into the document. These format-specific tables will only be included in the output if a document of the respective format is produced. This is method can be extended to apply any kind of format-specific typographic functionality which would otherwise be unavailable in Markdown syntax.

## Figures and images

Images are inserted as follows:

```
![alt text](image location/ name)
```

e.g.,

```

```

The *alt text* is used e.g., in HTML output. Image dimensions can be defined in braces:

```
{width=50mm}
```

As well, an identifier for the figure can be defined with #, resulting e.g., in the image attributes `{#figure1 height=30%}`.

A paragraph containing only an image is interpreted as a figure. The *alt text* is then output as the figure's caption.

## Symbols

Scientific texts often require special characters; for example, Greek letters, mathematical and physical symbols, and so on.

The UTF-8 standard, developed and maintained by the *Unicode Consortium*, enables the use of characters across languages and computer platforms. The encoding is defined as RFC document 3629 of the Network Working group (*Yergeau, 2003*) and as ISO standard ISO/IEC 10646:2014 (*International Organization for Standardization, 2014*). Specifications of Unicode and code charts are provided on the Unicode homepage (http://www.unicode.org/).

In Pandoc Markdown documents, Unicode characters such as ∘, $\alpha$, ä, Å can be inserted directly and passed to the different output documents. The correct processing of MD with UTF-8 encoding to LATEX/PDF output requires the use of the `--latex-engine=xelatex` option and the use of an appropriate font. The Times-like XITS font (https://github.com/khaledhosny/xits-math), suitable for high quality typesetting of scientific texts, can be set in the LATEX template:

```
\usepackage{unicode-math}
\setmainfont
[    Extension = .otf,
   UprightFont = *-regular,
      BoldFont = *-bold,
    ItalicFont = *-italic,
BoldItalicFont = *-bolditalic,
]{xits}
\setmathfont
[    Extension = .otf,
      BoldFont = *bold,
]{xits-math}
```

To facilitate the input of specific characters, so-called mnemonics can be enabled in some editors (e.g., in atom by the `character-table` package). For example, the 2-character mnemonics ':u' gives 'ü' (diaeresis), or 'D*' the Greek Δ. The possible character mnemonics and character sets are listed in RFC 1345: http://www.faqs.org/rfcs/rfc1345.html (*Simonsen, 1992*).

## Formulas

Formulas are written in LATEX mode using the delimiters $. For example, the formula for calculating the standard deviation *s* of a random sampling would be written as:

`$s=\sqrt{\frac{1}{N-1}\sum_{i=1}^N(x_i-\overline{x})^{2}}$`

and gives:

$$s = \sqrt{\frac{1}{N-1}\sum_{i=1}^{N}(x_i - \overline{x})^2}$$

with $x_i$ the individual observations, $\overline{x}$ the sample mean and $N$ the total number of samples.

Pandoc parses formulas into internal structures and allows conversion into formats other than LATEX. This allows for format-specific formula representation and enables computational analysis of the formulas (*Corbí & Burgos, 2015*).

## Code listings

Verbatim code blocks are indicated by three tilde symbols:

```
~~~
verbatim code
~~~
```

Typesetting `inline code` is possible by enclosing text between back ticks.

```
`inline code`
```

## Other document elements

These examples are only a short demonstration of the capacities of Pandoc concerning scientific documents. For more detailed information, we refer to the official manual (http://pandoc.org/MANUAL.html).

# CITATIONS AND BIOGRAPHY

The efficient organization and typesetting of citations and bibliographies is crucial for academic writing. Pandoc supports various strategies for managing references. For processing the citations and the creation of the bibliography, the command line parameter `--filter pandoc-citeproc` is used, with variables for the reference database and the bibliography style. The bibliography will be located automatically at the header `# References` or `# Bibliography`.

## Reference databases

Pandoc is able to process all mainstream literature database formats, such as RIS, BIB, etc. However, for maintaining compatibility with LATEX/ BIBTEX, the use of BIB databases is recommended. The used database either can be defined in the YAML metablock of the MD file (see below) or it can be passed as parameter when calling Pandoc.

### Inserting citations

For inserting a reference, the database key is given within square brackets, and indicated by an '@'. It is also possible to add information, such as page:

```
[@suber_open_2012; @benkler_wealth_2006, 57 ff.]
```

gives (*Suber, 2012*; *Benkler, 2006*, p. 57 ff.).

### Styles

The Citation Style Language (CSL) (http://citationstyles.org/) is used for the citations and bibliographies. This file format is supported, for example, by the reference management programs Mendeley (https://www.mendeley.com/), Papers (http://papersapp.com/) and Zotero (https://www.zotero.org/). CSL styles for particular journals can be found from the Zotero style repository (https://www.zotero.org/styles). The bibliography style that Pandoc should use for the target document can be chosen in the YAML block of the Markdown document or can be passed in as an command line option. The latter is more recommendable, because distinct bibliography style may be used for different documents.

### Creation of LATEX `natbib` citations

For citations in scientific manuscripts written in LATEX, the natbib package is widely used. To create a LATEX output file with natbib citations, Pandoc simply has to be run with the `--natbib` option, but without the `--filter pandoc-citeproc` parameter.

### Database of cited references

To share the bibliography for a certain manuscript with co-authors or the publisher's production team, it is often desirable to generate a subset of a larger database, which only contains the cited references. If LATEX output was generated with the `--natbib` option, the compilation of the file with LATEX gives an AUX file (in the example named `md-article.aux`), which subsequently can be extracted using BibTool (https://github.com/ge-ne/bibtool):

```
~~~
bibtool -x md-article.aux -o bibshort.bib
~~~
```

In this example, the article database will be called `bibshort.bib`.

For the direct creation of an article specific BIB database without using LATEX, we wrote a simple Perl script called `mdbibexport` (https://github.com/robert-winkler/mdbibexport).

## META INFORMATION OF THE DOCUMENT

*Bourne (2005)* argues that journals should be effectively equivalent to biological databases: both provide data which can be referenced by unique identifiers like DOI or, for example, gene IDs. Applying the semantic-web ideas of *Berners-Lee & Hendler (2001)* to this domain can make this vision a reality. Here we show how metadata can be specified in Markdown. We propose conventions, and demonstrate their suitability to enable interlinked and semantically enriched journal articles.

Document information such as title, authors, abstract etc. can be defined in a metadata block written in YAML syntax. YAML ("YAML Ain't Markup Language", http://yaml.org/) is a data serialization standard in simple, human readable format. Variables defined in the YAML section are processed by Pandoc and integrated into the generated documents. The YAML metadata block is recognized by three hyphens (`---`) at the beginning, and three hyphens or dots (`...`) at the end; for example;

```
---
title: Formatting Open Science
subtitle: agile creation of multiple document types
date: 2017-02-10

...
```

The public availability of all relevant information is a central aspect of Open Science. Analogous to article contents, data should be accessible via default tools. We believe that this principle must also be applied to article metadata. Thus, we created a custom Pandoc writer that emits the article's data as JSON–LD (*Lanthaler & Gütl, 2012*), allowing for informational and navigational queries of the journal's data with standard tools of the semantic web. The above YAML information would be output as:

```
{
  "@context": {
    "@vocab": "http://schema.org/",
    "date": "datePublished",
    "title": "headline",
    "subtitle": "alternativeTitle"
  },
  "@type": "ScholarlyArticle",
  "title": "Formatting Open Science",
  "subtitle": "agile creation of multiple document types",
  "date": "2017-02-10"
}
```

This format allows processing of the information by standard data processing software and browsers.

## Flexible metadata authoring

We developed a method to allow writers the flexible specification of authors and their respective affiliations. Author names can be given as a string, via the key of a single-element object, or explicitly as a `name` attribute of an object. Affiliations can be specified directly as properties of the author object, or separately in the `institute` object.

Additional information, for example, email addresses or identifiers like ORCID (*Haak et al., 2012*), can be added as additional values:

```
author:
  - John Doe:
```

```
        institute: fs
        email: john.doe@example.com
        orcid: 0000-0000-0000-0000
institute:
  fs: Science Formatting Working Group
```

## JATS support

The journal article tag suite (JATS) was developed by the National Library of Medicine (NLM) and standardized by ANSI/NISO as an archiving and exchange format of journal articles and the associated metadata (*National Information Standards Organization, 2012*), including data of the type shown above. The `pandoc-jats` writer by Martin Fenner is a plugin usable with Pandoc to produce JATS-formatted output. The writer was adapted to be compatible with our metadata authoring method, allowing for simple generation of files which contain the relevant metadata.

## Citation types

Writers can add information about the reason a citation is given. This might help reviewers and readers, and can simplify the search for relevant literature. We developed an extended citation syntax that integrates seamlessly into Markdown and can be used to add complementary information to citations. Our method is based on CiTO, the Citation Typing Ontology (*Shotton, 2010*), which specifies a vocabulary for the motivation when citing a resource. The type of a citations can be added to a Markdown citation using `@CITO_PROPERTY:KEY`, where `CITO_PROPERTY` is a supported CiTO property, and `KEY` is the usual citation key. Our tool extracts that information and includes it in the generated linked data output. A general CiTO property (*cites*) is used, if no CiTO property is found in a citation key.

The work at hand will always be the subject of the generated semantic *subject–predicate–object* triples. Some CiTO predicates cannot be used in a sensical way under this condition. Focusing on author convenience, we use this fact to allow shortening of properties when sensible. For example, if authors of a biological paper include a reference to the paper describing a method which was used in their work, this relation can be described by the *uses_method_in* property of the CiTO ontology. The inverse property, *provides_method_for*, would always be nonsensical in this context as implied by causality. It is therefore not supported by our tool. This allows us to introduce an abbreviation (*method*) for the latter property, as any ambiguity has been eliminated. Users of Western blotting might hence write `@method_in:towbin_1979` or even just `@method:towbin_1979`, where *towbin_1979* is the citation identifier of the describing paper by *Towbin, Staehelin & Gordon (1979)*.

## EXAMPLE: MANUSCRIPT WITH OUTPUT OF DOCX/ODT FORMAT AND LATEX/PDF FOR SUBMISSION TO DIFFERENT JOURNALS

Scientific manuscripts have to be submitted in a format defined by the journal or publisher. At the moment, DOCX is the most common file format for manuscript submission. Some

publishers also accept or require LATEX or ODT formats. Additional to the general style of the manuscript—organization of sections, fonts, etc.—the citation style of the journal must also be followed. Often, the same manuscript has to be prepared for different journals; for example, if the manuscript was rejected by a journal and has to be formatted for another one, or if a preprint of the paper is submitted to an archive that requires a distinct document format than the targeted peer-reviewed journal. In this example, we want to create a manuscript for a *PLoS* journal in DOCX and ODT format for WYSIWYG word processors. Further, a version in LATEX/ PDF should be produced for PeerJ submission and archiving at the PeerJ preprint server.

The examples for DOCX/ ODT are kept relatively simple, to show the proof-of-principle and to provide a plain document for the development of own templates. Nevertheless, the generated documents should be suitable for submission after little manual editing. For specific journals it may be necessary to create more sophisticated templates or to copy/ paste the generic DOCX/ ODT output into the publisher's template.

### Development of a DOCX/ ODT template

A first DOCX document with bibliography in *PLoS* format is created with Pandoc DOCX output:

```
pandoc -S -s --csl=plos.csl --filter pandoc-citeproc
  -o pandoc-manuscript.docx agile-editing-pandoc.md
```

The parameters `-S -s` generate a typographically correct (dashes, non-breaking spaces etc.) stand-alone document. A bibliography with the *PLoS* style is created by the citeproc filter setting `--csl=plos.csl --filter pandoc-citeproc`.

The document settings and styles of the resulting file `pandoc-manuscript.docx` can be optimized and be used again as document template (`--reference-docx=pandoc-manuscript.docx`): instead of .

```
pandoc -S -s --reference-docx=pandoc-manuscript.docx --csl=plos.csl
  --filter pandoc-citeproc -o outfile.docx agile-editing-pandoc.md
```

It is also possible to directly re-use a previous output file as template (i.e., template and output file have the same file name):

```
pandoc -S -s --columns=10 --reference-docx=pandoc-manuscript.docx
  --csl=plos.csl --filter=pandoc-citeproc
  -o pandoc-manuscript.docx agile-editing-pandoc.md
```

In this way, the template can be incrementally adjusted to the desired document formatting. The final document may be employed later as Pandoc template for other manuscripts with the same specifications. In this case, running Pandoc the first time with the template, the contents of the new manuscript would be filled into the provided DOCX template. A page with DOCX manuscript formatting of this article is shown in Fig. 8.

The same procedure can be applied with an ODT formatted document.

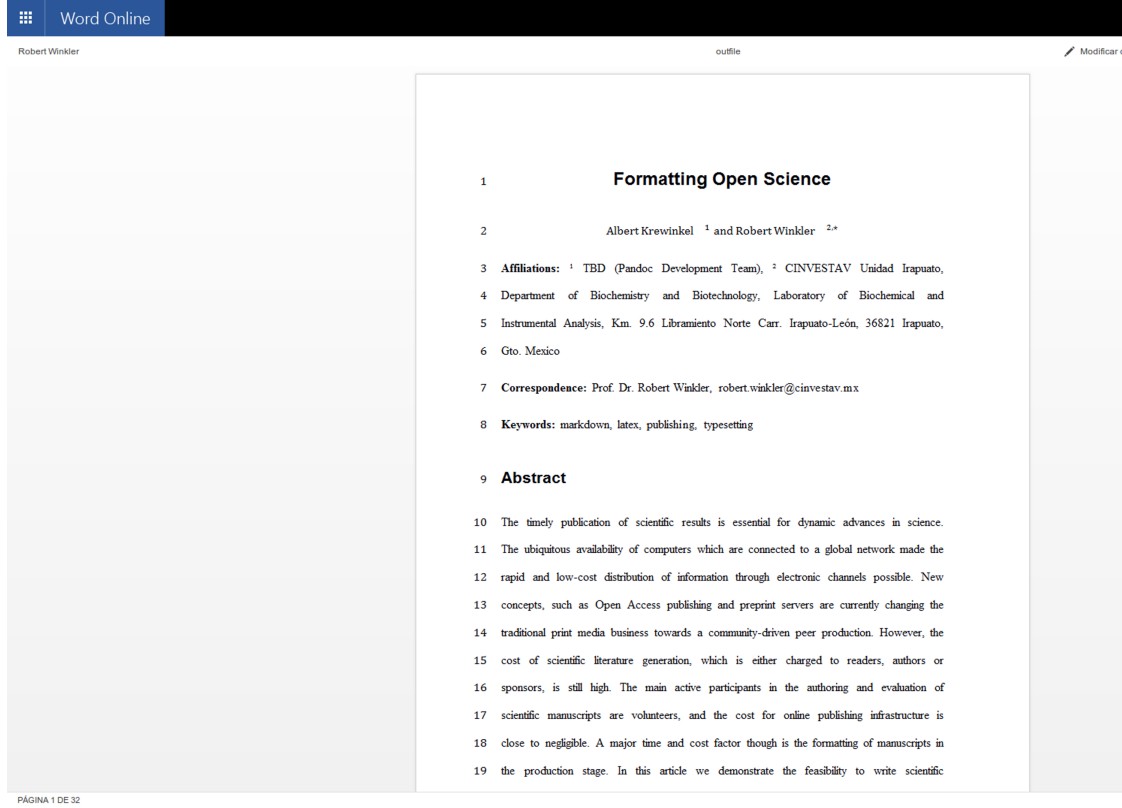

**Figure 8** Opening a Pandoc-generated DOCX in Microsoft Office 365.

## Development of a TEX/PDF template

The default Pandoc LATEX template can be written into a separate file by:

```
pandoc -D latex > template-peerj.latex
```

This template can be adjusted; for example, by defining Unicode encoding (see above), by including particular packages or setting document options (line numbering, font size). The template can then be used with the Pandoc parameter `--template=pandoc-peerj.latex`.

The templates used for this document are included as Supplemental Material (see section '*Software and Code Availability*' below).

## Styles for HTML and EPUB

The style for HTML and EPUB formats can be defined in .css stylesheets. The Supplemental Material (see section 'Software and Code Availability' below) contains a simple example .css file for modifying the HTML output, which can be used with the Pandoc parameter `-c pandoc.css`.

## AUTOMATING DOCUMENT PRODUCTION

The commands necessary to produce the document in a specific formats or styles can be defined in a simple `Makefile`. An example `Makefile` is included in the source code of

**Table 3  Relevant software used for this article.**

| Software | Use | Authors | Version | Release | Homepage/repository |
|---|---|---|---|---|---|
| Pandoc | Universal markup converter | John MacFarlane | 1.16.0.2 | 16/01/13 | http://www.pandoc.org |
| Pandoc-citeproc | Library for CSL citations with Pandoc | John MacFarlane, Andrea Rossato | 0.9.1 | 16/03/19 | https://github.com/jgm/pandoc-citeproc |
| Pandoc-jats | Creation of JATS files with Pandoc | Martin Fenner | 0.9 | 15/04/26 | https://github.com/mfenner/pandoc-jats |
| ownCloud | Personal cloud software | ownCloud GmbH, Community | 9.1.1 | 16/09/20 | https://owncloud.org/ |
| Markdown Editor | Plugin for ownCloud | Robin Appelman | 0.1 | 16/03/08 | https://github.com/icewind1991/files_markdown |
| BibTool | Bibtex database tool | Gerd Neugebauer | 2.63 | 16/01/16 | https://github.com/ge-ne/bibtool |

this article. The desired output file format can be chosen when calling make. For example, make outfile.pdf produces this article in PDF format. Calling make without any option creates all listed document types. A Makefile producing DOCX, ODT, JATS, PDF, LATEX, HTML and EPUB files of this document is provided as Supplemental Material (see section 'Software and Code Availability' below).

### Cross-platform compatibility

The make process was tested on Windows 10 and Linux 64 bit. All documents—DOCX, ODT, JATS, LATEX, PDF, EPUB and HTML—were generated successfully, which demonstrates the cross-platform compatibility of the workflow.

## PERSPECTIVE

Following the trend to peer production, the formatting of scientific content must become more efficient. Markdown/Pandoc has the potential to play a key role in the transition from proprietary to community-driven academic production. Important research tools, such as the statistical computing and graphics language R (*R Core Team, 2014*) and the Jupyter notebook project (*Kluyver et al., 2016*) have already adopted the MD syntax (for example, http://rmarkdown.rstudio.com/). The software for writing manuscripts in MD is mature enough to be used by academic writers. Therefore, publishers also should consider implementing the MD format into their editorial platforms.

## CONCLUSIONS

Authoring scientific manuscripts in markdown (MD) format is straight-forward, and manual formatting is reduced to a minimum. The simple syntax of MD facilitates document editing and collaborative writing. The rapid conversion of MD to multiple formats such as DOCX, LATEX, PDF, EPUB and HTML can be done easily using Pandoc, and templates enable the automated generation of documents according to specific journal styles.

The additional features we implemented facilitate the correct indexing of meta information of journal articles according to the 'semantic web' philosophy.

Altogether, the MD format supports the agile writing and fast production of scientific literature. The associated time and cost reduction especially favours community-driven publication strategies.

## SOFTWARE AND CODE AVAILABILITY

The relevant software for creating this manuscript used is cited according to (*Smith, Katz & Niemeyer, 2016*) and listed in Table 3. Since unique identifiers are missing for most software projects, we only refer to the project homepages or software repositories:

## ACKNOWLEDGEMENTS

We cordially thank Dr. Gerd Neugebauer for his help in creating a subset of a bibtex data base using BibTool, as well as Dr. Ricardo A. Chávez Montes, Prof. Magnus Palmblad and Martin Fenner for comments on the manuscript. Warm thanks also go to Anubhav Kumar and Jennifer König for proofreading.

### Funding

The work was funded by the Consejo Nacional de Ciencia y Tecnología (CONACyT) Mexico, with the grant FRONTERAS 2015-2/814 and by institutional funding of the Centro de Investigación y de Estudios Avanzados del Instituto Politécnico Nacional (CINVESTAV). The funders had no role in study design, data collection and analysis, decision to publish, or preparation of the manuscript.

### Grant Disclosures

The following grant information was disclosed by the authors:
Consejo Nacional de Ciencia y Tecnología (CONACyT).
FRONTERAS 2015-2/814.
Centro de Investigación y de Estudios Avanzados del Instituto Politécnico Nacional (CINVESTAV).

### Competing Interests

Albert Krewinkel is a voluntary member of the Pandoc Development Team. Robert Winkler is an Academic Editor for PeerJ. We have no financial/ legal conflict of interest.

### Author Contributions

- Albert Krewinkel and Robert Winkler conceived and designed the experiments, performed the experiments, analyzed the data, contributed reagents/materials/analysis tools, wrote the paper, prepared figures and/or tables, performed the computation work, reviewed drafts of the paper, programming.

## Data Availability

The software created as part of this article, Pandoc-scholar, is suitable for general use and has been published at https://github.com/pandoc-scholar/pandoc-scholar; 10.5281/zenodo.376761. The source code of this manuscript, as well as the templates and Pandoc Makefile, have been deposited to https://github.com/robert-winkler/scientific-articles-markdown/.

Drawings for document types, devices and applications have been adopted from Calibre (http://calibre-ebook.com/), openclipart (https://openclipart.org/) and the GNOME Theme Faenza (https://code.google.com/archive/p/faenza-icon-theme/).

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

## FURTHER READING

**DPT Collective. 2015.** Monk J, Rasch M, Cramer F, Wu A, eds. *From print to ebooks: a hybrid publishing toolkit for the arts.* Amsterdam: Institute of Network Cultures.