# Peer review of "Formatting Open Science: agilely creating multiple document formats for academic manuscripts with Pandoc Scholar"

_PeerJ Computer Science, doi:10.7717/peerj-cs.112_

## Round 0.1 · original submission · Major Revisions

Please follow the recommendations of both reviewers, specially the first one, who had more serious concerns about the quality of the paper.

Reviewer 1 ·

Basic reporting

The manuscript does not follow the traditional structure of a research article but is still easy to follow. However, I consider that it would benefit a more traditional IMRAD approach since that would highlight the actual individual parts (and contribution) of the manuscript better than the current custom format where parts are more blended together.

I think the literature review concerning Open Access and article processing fees is only tangentially relevant to the article, would consider de-emphasizing that and rather expanding the review of literature that concerns the different markup languages (which I see as the main contribution of this article).

Language is of good quality but should further be improved, counted a handful of simple mistakes during my time with the manuscript.

The discussion section is non-existent, there needs to be ties back into both existing research as well as describing avenues for how this work can be used as an outgoing step for future research.

Experimental design

There is not really a research question in the traditional sense and that I think is the main problem - the manuscript needs to be written more in the style of an academic article rather than a more casual guide for how-to guide for using markdown.

I am not fully convinced that this article fulfils the criteria of being a 'research article' since f markdown and pandoc are presented largely in a vacuum from the section "CONCEPTS OF MARKDOWN AND PANDOC" onwards.

Could some kind of evaluation criteria be introduced to compare the different markup languages to each other to really emphasise the benefits of markdown? I think something like this would be needed.

Validity of the findings

The application of markdown in the context of academic writing warrants focus and I think it is a worthy pursuit to conduct research around it. However, the current manuscript does not fulfil some of the basic criteria for a research-based article but includes to much anecdotal,unquestioned, or unevaluated solutions. I would like to read more about the benefits of markdown vs. LATEX other than being more simple in its syntax.

Additional comments

I am interested in seeing how you may improve on the manuscript since I see potential in the topic, however, the current packaging has a lot that can and should be improved before publication.

Reviewer 2 ·

Basic reporting

This paper describes a strategy for manuscript writing that facilitates the creation of various output documents. The work is well-written and technically sound. Related work is enough detailed, introducing Open Access and justifying the motivation of their work.

In my opinion, some improvements could be introduced.

- Some figures could be resized: Figure 1, 3 and 4 could be smaller. Figure 5, 6 and 7 could be bigger.
- Table 2 may have some error. Tags h1 and h2 may have an additional "<" and ">". Example of bold typeface in HTML could have unnecessary elements (**).
- Some elements of Figure 3 could be better explained. Why MD is linked with bib? In this very figure, template is included in pandoc but no explication is given about it.
- Figure 6 includes an example of owncloud, however, a reader who does not know this platform could not understand what it is. Maybe, a very little introduction could be helpful.

Experimental design

The proposed methodology is compared with others well-known methods and some examples are depicted.

However, some changes could be included:

- Example of how to define additional parameters in figures should be included (Page 10, Line 244).
- In subsection Symbols (Page 10) an example of latex is included. In my opinion, it is not clear if the authors are comparing with latex or this example is the instruction in case of you want to transform MD to Latex.
- Section of "Manuscript with output of DOCX/ODT format and LATEX/PDF for submission to different journals" should be better explained. The listed command could be described. In my opinion, the aim of the section and the commands should be better defined.
The use of templates is interesting but it is not well-defined.

Validity of the findings

Conclusions are clear and well stated.

Additional comments

The use of templates are one of the most interesting parts of Pandoc, but almost no information is provided. This is the weak link of the article.

---

## Round 0.2 · accepted · Accept

Both reviewers consider the paper is ready for publication and so do I.

Reviewer 1 ·

Basic reporting

The authors have adequately responded to my earlier comments and improved the manuscript accordingly or explained why certain things are as they are in manuscript, I now see no major problems with the basic reporting other than that the discussion section and connections back to existing research could still be improved. However, I do not see this as a critical omission that would pose an obstacle for publishing the article.

Experimental design

The authors have adequately responded to my earlier comments and improved the manuscript accordingly.

Validity of the findings

The authors have adequately responded to my earlier comments and improved the manuscript accordingly.

Additional comments

Good work on improving the manuscript While I do recommend accepting the article I still think there is fairly easy work that could be done to make it more contextualized to existing research particularly in the last few pages of the manuscript, now the results don“t really feed back into any existing streams of research despite being based on a plethora of references at the initial outset.

Reviewer 2 ·

Basic reporting

This paper describes a strategy for manuscript writing that facilitates the creation of various output documents. The work is well-written and technically sound. Related work is enough detailed, introducing Open Access and justifying the motivation of their work.

However, one previous minor modification asked in the last review remains unchanged. According to my understanding, Table 2 may continue having one error. Tags h1 and h2 may have an additional "<" and ">".

Apart from that, I recommend to accept the manuscript. Congratulations for the authors.

Experimental design

Experimental design has been improved with regard to the previous version, and new interesting features have been included.

Validity of the findings

Conclusions are clear and well stated.